# Therapeutic interventions on human breast cancer xenografts promote systemic dissemination of oncogenes

**Gorantla V. Raghuram**[1,2☯], **Kavita Pal**[1,2☯], **Gaurav Sriram**[1,2], **Afzal Khan**[1,2], **Ruchi Joshi**[1,2], **Vishalkumar Jadhav**[1,2], **Sushma Shinde**[1,2], **Alfina Shaikh**[1,2], **Bhagyeshri Rane**[1,2], **Harshada Kangne**[1,2], **Indraneel Mittra**[1,2]*

1 Translational Research Laboratory, Tata Memorial Centre, Advanced Centre for Treatment, Research and Education in Cancer, Navi Mumbai, India, 2 Homi Bhabha National Institute, Mumbai, India

☯ These authors contributed equally to this work.
* imittra@actrec.gov.in

**Data Availability Statement:** All relevant data are within the paper.

**Funding:** This study was supported by the Department of Atomic Energy, Government of

## Abstract

Metastatic dissemination following successful treatment of the primary tumour remains a common cause of death. There is mounting evidence that therapeutic interventions themselves may promote development of metastatic disease. We earlier reported that cell-free chromatin particles (cfChPs) released from dying cancer cells are potentially oncogenic. Based on this observation we hypothesized that therapeutic interventions may lead to the release of cfChPs from therapy induced dying cancer cells which could be carried via the blood stream to distant organs to transform healthy cells into new cancers that would masquerade as metastasis. To test this hypothesis, we generated xenografts of MDA-MB-231 human breast cancer cells in severe combined immune-deficient mice, and using immunofluorescence and FISH analysis looked for cfChPs in their brain cells. We detected multiple human DNA signals representing cfChPs in nuclei of brain cells of mice which co-localized with eight human onco-proteins. No intact MDA-MB-231 cells were detected. The number of co-localizing human DNA and human c-Myc signals increased dramatically following treatment with chemotherapy, localized radiotherapy or surgery, which could be prevented by concurrent treatment with three different cfChPs deactivating agents. These results suggest that therapeutic interventions lead to the release cfChPs from therapy induced dying cancer cells carrying oncogenes and are transported via the blood stream to brain cells to potentially transform them to generate new cancers that would appear as metastases. cfChPs induced metastatic spread of cancer is preventable by concurrent treatment with agents that deactivate cfChPs.

## Introduction

The most common cause of cancer-related mortality is metastasis. The latter is a complex phenomenon and despite decades of intensive research the processes behind the various steps involved in the metastatic cascade are not well understood [1, 2]. This scenario is especially

India, through its grant CTCTMC to Tata Memorial Centre awarded to IM. The funder had no role in the preparation, review, or approval of the manuscript and decision to submit the manuscript for publication.

**Competing interests:** The authors declare no competing interests.

disappointing since metastatic dissemination following successful treatment of the primary tumour remains a common cause of death. There is mounting evidence that therapeutic interventions themselves may promote metastatic spread of cancer [3]. This possibility has been raised with respect to all three modalities of cancer treatment viz. chemotherapy [4, 5], radiotherapy [6, 7] and surgery [8, 9]. We earlier reported that cell-free chromatin particles (cfChPs) that are released from dying cancer cells can integrate into genomes of healthy bystander cells to activate two critical hallmarks of cancer, namely, dsDNA breaks and inflammation [10]. In keeping with the classical report of Isaiah Fidler that 99% of intravenously injected cancer cells die by 24 h [11], we observed that when injected intravenously, the vast majority of cancer cells died to release cfChPs which integrated into the genomes of brain, lung and liver cells of mice [10]. This resulted in dsDNA breaks marked by phosphorylation of H2AX, and activation of the pro-inflammatory transcription factor NFκB and the inflammatory cytokines, IL-6, TNFα and IFNγ. Since concurrent activation of DNA damage and inflammation are potent stimuli of oncogenic transformation [12, 13], we hypothesized that cfChPs released from therapy induced dying cancer cells might be carried via the blood stream to distant organs to oncogenically transform the target cells to generate new cancers which would masquerade as metastasis [10]. We tested this hypothesis in a pre-clinical model using MDA-MB-231 human breast cancer cell xenografts in severe combined immune-deficient (SCID) mice and found that therapeutic interventions viz. chemotherapy, radiotherapy and surgical excision of the xenograft led to systemic dissemination of cfChPs carrying eight human oncogenes which accumulated in their brain cells. We did not detect any intact MDA-MB-231 cells. We postulate that the oncogenes contained within cfChPs that are released from therapy induced dying cancer cells and transported via blood stream to the brain could potentially transform brain cells to generate new cancers that would masquerade as metastasis. Such a model of cancer metastasis would pose a challenge to the conventional model which proposes that cells released from the primary tumour are themselves carried to distant organs to grow and generate metastasis.

## Materials and methods

### Animal ethics approval

The experimental protocol of this study was approved by the Institutional Animal Ethics Committee (IAEC) of the Advanced Centre for Treatment, Research and Education in Cancer (ACTREC), Tata Memorial Centre (TMC), Navi Mumbai, India under permission number 32/2016. The experiments were carried out in compliance with the ethical regulations and humane endpoint criteria of IAEC and ARRIVE guidelines. The Plos One checklist of ARRIVE guidelines is given in S2 Table.

ACTREC- IAEC maintains respectful treatment and care of animals in scientific research. It aims that the use of animals in research contributes to the advancement of knowledge following ethical and scientific necessities. All scientists and technicians involved in this study have undergone training in ethical handling and management of animals under supervision of FELASA certified attending veterinarians. They affirm that respect for all forms of life is an inherent characteristic of biological and medical scientists who conduct research involving animals. Animals were euthanized under sterile conditions at appropriate time points under $CO_2$ atmosphere by cervical dislocation under supervision of FELASA trained animal facility personnel.

All welfare considerations were taken to minimize any suffering and distress due to creation of the tumour xenografts. While considering humane end points, it was ensured that xenografts did not reach a size of 10 mm in any one dimension. In addition, activity and weight loss

of more than 15% were considered as humane end points. Mice were observed for the above humane end points once every two days. None of the animals were found to have reached the above humane end points at the time of sacrifice. None of the animals died during the total duration of experiment of ~48 days.

The PLOS ONE Humane Endpoints Checklist is given as S3 Table.

## SCID mice

Inbred female NOD SCID (NOD.Cg-Prkdcscid/J) mice were used in this study. They were obtained from the institutional animal facility and were maintained according to IAEC standards. They were housed under maximum-barrier facilities in sterilized cages with filter tops and sterile bedding material. The mice were fed γ-irradiated sterilized food and water ad libitum and kept under 12-h light/dark cycle with free access to water and food. A HVAC system was used to provide controlled room temperature, humidity and air pressure.

## Creation of xenografts

NOD SCID mice (6–8 week old) were inoculated under the lower dorsal skin with $1 \times 10^6$ MDA-MB-231 human breast cancer cells and xenografts were allowed to grow to a size of ~0.125 cm$^3$, which usually took ~6 weeks, when the experiments were initiated (S1 Fig).

## Therapeutic interventions

**Chemotherapy.** Mice were administered a single i.p. injection of Adriamycin, 0.65 mg/kg in 100μl of saline.

**Tumour irradiation.** Mice were anesthetized using ketamine (80 mg/kg IP) and xylazine (10 mg/kg IP) and placed in a polypropylene box. The box was placed on the couch of a telecobalt machine (Bhabatron-II)$^©$ and positioned in such a way that the apex of the tumor was at the center of the aperture of the machine. The rest of the body of the mice was shielded from radiation using 6.5 cm-thick lead shields (S2 Fig). A dose of radiation was delivered to the tumors (10Gy).

**Surgical excision.** Mice were anesthetized using ketamine (80 mg/kg IP) and xylazine (10 mg/kg IP) and the xenografts were surgically excised.

## cfChPs deactivating agents

The anti-cancer therapies were given either alone or in conjunction with three different cfChPs deactivating agents viz. anti-histone antibody complexed nanoparticles (CNPs) [14], DNase I or a combination of the nutraceuticals Resveratrol and copper (R-Cu) [15, 16]. Combining Resveratrol (R) with copper (Cu) leads to the generation of free radicals [17] which can effectively deactivate cfChPs by degrading their DNA components [15, 16].

**Anti-histone antibody complexed nanoparticles (CNPs).** CNPs were prepared according to the method reported by us earlier [14] except that histone H4 IgG was exclusively used for preparing CNPs. CNPs, 50 μg in 150 μl PBS, were administered once a day for 5 days.

**DNase I.** DNase I (Sigma-Aldrich; Catalogue No- DN25-1G) dissolved in saline was administered twice daily at a dose of 15 mg/kg in 100 μl of saline i.p.

**Resveratrol-copper (R-Cu).** R: 1 mg/Kg in 100μl water, and Cu: $10^{-4}$ mg/Kg in 100μl water, were administered by oral gavage one after the other twice daily. Resveratrol (Trade name—TransMaxTR) was sourced from Biotivia LLC, USA; Copper (Trade name—Chelated Copper) was sourced from J.R. Carlson Laboratories Inc. USA.

## Study design

The study design is depicted in the schema given in S3 Fig.

Mice were sacrificed under $CO_2$ atmosphere by cervical dislocation under supervision of FELASA trained animal facility personnel 5 days (i.e. on day 6) after starting anti-cancer treatments; cfChPs deactivating agents were started 4h prior to commensing anti-cancer therapies. Brains of mice were removed, fixed in formalin and FFPE sections were prepared for analysis.

**Experiment no.1.** Involve XG bearing mice without anti-cancer treatments. Their brain sections were analyzed for detection of human DNA and human onco-proteins by immuno-FISH.

**Experiment no.2.** Involve XG bearing mice receiving anti-cancer treatments. Statistical comparison was performed on the effects of anti-cancer treatments, with and without cfChPs deactivating agents, on human DNA and c-Myc onco-protein signals in brains cells by immuno-FISH.

## Immunofluorescence and FISH analysis

The FFPE sections of the brain were analyzed by immuno-FISH using a human specific whole genomic probe and specific antibodies against human HLA-ABC antigen and eight different human oncogenes viz. c-Myc; c-Raf, p-EGFR, HRAS, p-AKT, FGFR 3, PDGFRA and c-Abl. For FISH analysis, 500 cells were examined for detection of human DNA signals (at a magnification of X60) and percentage of cells showing positive signals was calculated. For onco-protein analysis, 1000 cells were examined for detection of human c-Myc signals (at a magnification of X40) and percentage of cells showing positive signals for human c-Myc protein was calculated. Details of the FISH probe and various antibodies used in this study are provided in the S1 Table.

## Fluorescent dual-labelling of MDA-MB-231 cells and i.v. injection into SCID mice

The MDA-MB-231 cells were dually labelled in their DNA and in their histones according to a method described by us earlier [18]. MDA-MB-231 cells were plated at a density of $6 \times 10^4$ cells in Dulbecco's Modified Eagle Medium, and after overnight culture (cell count $1 \times 10^5$). DNA labelling was done using BrdU (10 μM / 1.5 ml for 24 h) and histone labelling was done using CellLight® Histone 2BGFP (60 μL / 1.5 ml for 36 h). BrdU was obtained from Sigma Chemicals, St. Louis, MO, USA; Catalogue No. B5002; CellLight® Histone 2BGFP was obtained from Thermo Fisher Scientific, Waltham MA, USA; Catalogue No. C10594. Representative images of the fluorescently double-labelled MDA-MB-231 cells are shown in S4a Fig. One hundred thousand dually labelled cells were injected via tail veins of two SCID mice and animals were killed after 72 h. Their brains were harvested and unstained FFPE sections were processed for fluorescence microscopy. Sections were pre-treated with anti-BrdU antibody before microscopy (Abcam, UK, Catalogue no. ab6326).

## Statistical analysis

Statistical comparison between the xenograft bearing group and the two control groups, and that between the xenograft bearing group and the three anti-cancer treatment groups (CT, RT and Sx) was done by two-tailed student t-test using GraphPad Prism 6.0 (https://www.graphpad.com/, GraphPad Software, Inc., USA). Statistical comparison between the three treatment groups (CT, RT and Sx) and those additionally treated with the three cfChPs deactivating agents was performed by One-way ANOVA using the same software.

## Results

Using immuno-FISH analysis on brain cells of xenograft bearing mice, we detected multiple co-localizing signals of human DNA and HLA-ABC in their brain cells (Fig 1a). Since the HLA-ABC antigen is unique to humans, and does not exist in mice, this finding provided strong support for the conclusion that DNA fragments carrying the HLA-ABC gene had been released from the dying human xenograft cells and had migrated to mouse brain cells via circulation. We also detected co-localizing signals of human DNA and eight human onco-proteins that we examined in brain cells of mice. The onco-proteins included c-Myc; c-Raf, p-EGFR, HRAS, p-AKT, FGFR 3, PDGFRA and c-Abl (Fig 1b). This finding indicated that the respective oncogenes had also been released from dying xenograft cells and carried via the blood stream to brain cells wherein they had expressed their respective proteins.

We next undertook a quantitative comparative analysis of human DNA and c-Myc onco-protein signals in brain cells following three anti-cancer interventions viz. a single intraperito-neal injection of Adriamycin, a single shot of localized radiation or surgical excision of the xenograft. These therapeutic interventions were given either alone or concurrently with three different cfChPs deactivating agents, namely, anti-histone antibody complexed nanoparticles (CNPs), DNase 1 and a combination of Resveratrol and copper (Fig 2a and 2b). The analysis was done in a blinded fashion such that the examiner was unaware of the group to which the immune-FISH slides belonged. As expected, no human DNA or c-Myc signals were detected in brain cells of mice in the two control groups viz. control mice without xenograft and control mice without xenograft but receiving anti-cancer interventions (Fig 2a and 2b). However, xenograft bearing mice (without therapeutic interventions) showed that a significant number of brain cell nuclei harbored human DNA (13.73%) and c-Myc onco-protein (3.70%) signals. This data indicated that dying cells of the xenografts had released their DNA carrying the c-Myc oncogene into the circulation which had migrated to the brain during the xenograft's growth span of ~6 weeks. The number of human DNA and c-Myc signals increased markedly following the three therapeutic interventions. With respect to DNA, the maximum increase in signals was seen with CT (13.73% vs 26.33%, $p < 0.001$), followed by RT (13.73% vs 21.83%, $p < 0.01$), followed by Sx (13.73% vs 15.83%, $p < 0.05$). With respect to c-Myc, the number of signals for CT were 3.70% vs 8.08% ($p < 0.01$) and for RT were 3.70% vs 9.33% ($p < 0.001$). However, for Sx, the difference in number of c-Myc signals in untreated and treated groups was not statistically significant (3.70% vs 3.37%). Concurrent treatment of mice with all three cfChPs deactivating agents showed remarkable and statistically significant reduction in both human DNA and c-Myc signals with p values ranging between $< 0.05$ and $< 0.01$ (Fig 2a and 2b).

Finally, we performed an experiment to doubly confirm that the fluorescent signals that we detected in the brain cells were not representative of intact metastatic cells that might have been released from the xenografts and had travelled to the brain. To this end, we dually fluo-rescently labelled MDA-MB-231 cells in their DNA and in their histones and injected them intravenously into SCID mice (S4a Fig). When sections of brain were examined after 72 h, we did not detect any intact dually fluorescently labelled cells representing metastasis; rather, what we found were dually labelled fluorescent particles representing cfChPs that had been released from dying xenograft cells (S4b Fig). This finding is in keeping with the observation of Fidler that 99% of cancer cells die within 24 h of intravenous injection [11].

## Discussion

Although systemic dissemination following successful treatment of the primary tumour is a common cause of death from the disease, an explanation for this paradox has remained elusive

Fig. 1

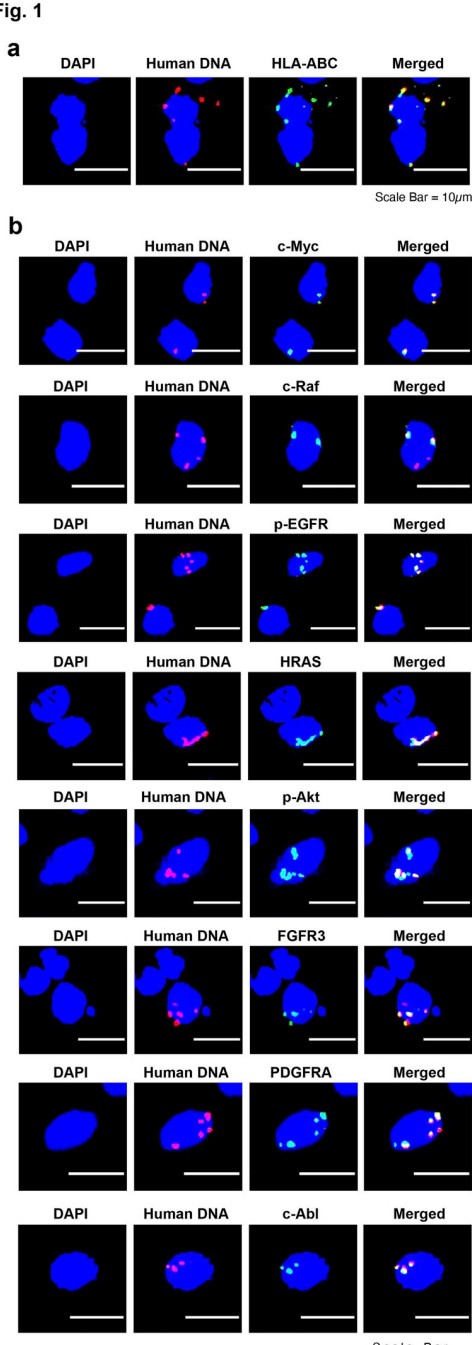

**Fig 1. Representative immuno-FISH images of FFPE sections of brains of mice bearing human breast cancer xenografts (without therapeutic interventions) showing co-localizing signals of human DNA and various human onco-proteins. a**. Co-localizing signals of human DNA and human HLA-ABC. **b**. Co-localizing signals of human DNA and eight different human onco-proteins.

[3]. We have shown here that cfChPs released from therapy induced dying cancer cells, and carrying oncogenes with them, can enter into the systemic circulation and be carried to distant organs such as the brain with the potential to oncogenically transform the brain cells to generate new cancers that would appear as metastasis. Such a scenario would provide a radically

Fig. 2

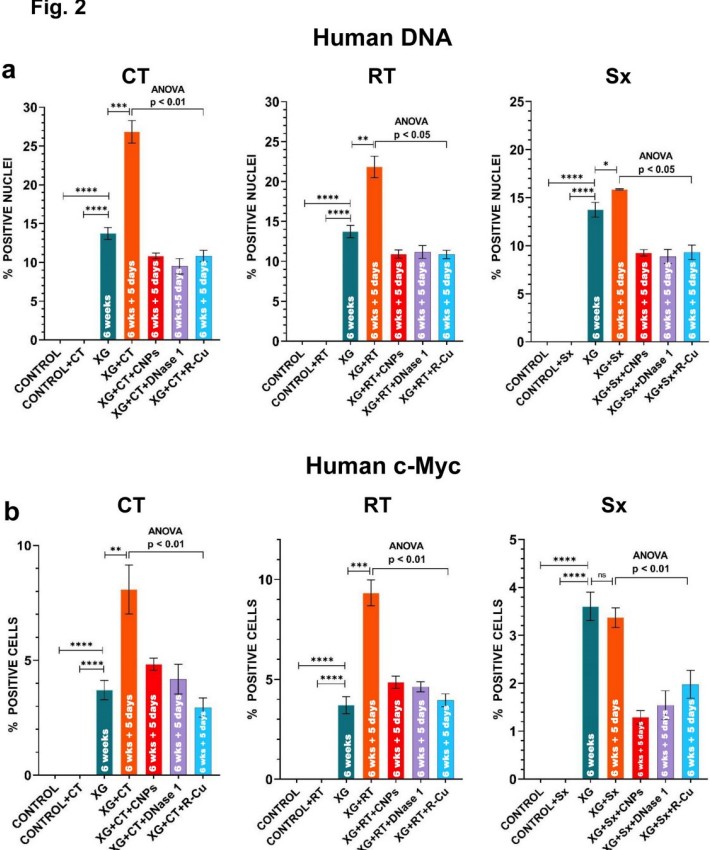

**Fig 2. Quantitative analyses depicted as histograms to illustrate that therapeutic interventions promote systemic dissemination of human DNA and human c-Myc oncoprotein to mouse brain cells, and that these can be prevented by concurrent treatment with three different cfChPs deactivating agents.** All groups had 4 mice each. **a**. Detection of human DNA by FISH **b**. Detection of human c-Myc onco-protein by immunofluorescence. Statistical comparison between the xenograft bearing group and the two control groups, and that between the xenograft bearing group and the three anticancer treatment groups (CT, RT and Sx) was done by two-tailed student t-test. * < 0.05, ** < 0.01, *** <0.001, **** <0.0001. Statistical comparison between the three anti-cancer treatment groups (CT, RT and Sx) and those additionally treated with the three cfChPs deactivating agents was done by One-way ANOVA. * < 0.05, ** < 0.01.

different model for cancer metastasis, wherein metastases arise de novo as new cancers from the cells of target organs transformed by cfChPs released from dying circulating cancer cells. The latter model would find mechanistic support from our earlier work which showed that cfChPs that circulate in blood of cancer patients can readily enter into cells of distant organs to damage their DNA and potentially transform them into new cancers [19].

The possibility that the fluorescence signals seen in Fig 1 represented intact MDA-MB-231 cells, as would be expected according to the conventional model of cancer metastasis, was excluded by our experiment wherein we intravenously injected dually fluorescently labelled MDA-MB-231 cells and looked for intact fluorescent cells in the brains of mice. Intact cells would be expected to appear as large dually-labelled fluorescent signals which would co-localize with an entire DAPI nuclear signal. We did not find any intact fluorescent cells; instead we detected fluorescent particles representing cfChPs within the blue DAPI nuclear signals (S4 Fig). Admittedly, there was a difference in timeframe between the above experiment in which

the animals were sacrificed at 72 h and those represented in Fig 1 wherein they were sacrificed at 6 weeks. However, if no intact MDA-MB-231 cells in the brain were seen at 72 h, it is highly unlikely that the fluorescent signals seen at 6 weeks would represent intact MDA-MB-231 cells. In fact, the absence of intact cells at 72 h is in keeping with Fidler's finding that the vast majority of the injected cancer cells rapidly die within 24 h [11].

Additionally, the fact that the fluorescent signals in brain cells could be markedly minimized by concurrent treatment with the three cfChPs deactivating agents strongly argues against the possibility that the fluorescent signals seen in brain cells represented intact MDA-MB-231 cells. Rather, they indicate that the vehicles that carried the c-Myc oncogene to brain cells were cfChPs released from therapy induced dying xenograft cells which could be prevented by deactivating cfChPs. This finding is consistent with our hypothesis that cfChPs are the critical oncogenic agents responsible for inducing metastasis [10].

There are several reports in the literature that membrane proteins can translocate to the nucleus [20, 21]. However, in our case, the oncogenes contained in the cfChPs that had accumulated in brain cells were being directly expressed as proteins in their nuclei. The relevance of such a finding is that oncogenes carried by cfChPs via the blood stream to the brain can potentially transform the brain cells to form new cancers which would appear as metastasis.

Whether therapy induced dissemination of oncogenes to brain cells would lead to development of metastases was not investigated in this short-term study. However, our recent report of a long-term experiment has confirmed such an eventuality by showing that intravenously injected MDA-MB-231 (and A375 human melanoma) cells died rapidly upon reaching the lung and released cfChPs that accumulated in the nuclei of lung cells [22]. The latter led to the activation of 10 hallmarks of cancer and the immune checkpoint PD-L1 in the lung cells as early as 72 h suggesting that the lung cells were rapidly transformed into incipient cancer cells. The lung metastases that subsequently developed after 2–3 months comprised of cells that were genetically chimeric and contained both mouse- and human-specific DNA, indicating that the cfChPs of human origin had integrated into the genomes of mouse lung cells and had transformed them into new cancers that developed as metastases. The ultimate confirmation that cfChPs can induce cancer metastasis came from the experiment in which purified cfChPs isolated from radiation killed MDA-MB-231 cells were intravenously injected into immune deficient mice which too generated lung metastasis. These findings has led us to conclude that metastasis arise de novo as new cancers from cells of target organs and not from those derived from cells of the primary tumour [22].

Our finding in this study that therapeutic interventions may potentially promote cancer metastasis poses a formidable challenge to the current policies of cancer treatment. Well-designed clinical trials need to be conducted to explore whether cfChPs deactivating agents given concurrently with anti-cancer treatments would prevent metastatic spread and improve survival of patients with cancer. Of the three cfChPs deactivating agents used in our study, the combination of Resveratrol and copper (R-Cu) would be the most attractive. Both resveratrol and copper are commonly used nutraceuticals and have been shown to be effective in multiple therapeutic indications in humans [23–26]. For instance, R-Cu treatment to patients with advanced oral cancer led to the down-regulation of ten hallmarks of cancer which included five immune check points that were being expressed by tumor-infiltrating lymphocytes [23]. If R-Cu is administered in conjunction with chemotherapy, it would also offer the added benefit of minimizing its toxic side effects [24, 25]. Taken together, these findings suggest that R-Cu may be a worthy candidate to be tested as an adjunct to cancer treatment in well-designed randomized clinical trials.

## Conclusion

Therapeutic interventions may potentially promote metastatic spread of cancer via the medium of cfChPs released from therapy induced dying cancer cells and carried via the blood stream to distant organs where they can oncogenecally transform healthy cells to generate new cancers which would masquerade as metastasis. The latter can be prevented by deactivating cfChPs using cfChPs deactivating agents concurrently with cancer treatments.

## Supporting information

**S1 Fig. A representative image of a SCID mouse bearing human breast cancer xenograft.**
(TIF)

**S2 Fig. Figure showing set-up for delivering localized radiation to xenografts (a and b).**
(TIF)

**S3 Fig. Schema of the study design.**
(TIF)

**S4 Fig. Figure showing fluorescently dual-labelled cfChPs in mouse brain (a and b).**
(TIF)

**S1 Table. Source of FISH probes and antibodies.**
(DOCX)

**S2 Table. ARRIVE guidelines.**
(DOCX)

**S3 Table. PLOS ONE Humane Endpoints Checklist.**
(DOCX)

## Acknowledgments

The authors thank Mr. Ashish Pawar for his help in preparing manuscript.

## Author Contributions

**Conceptualization:** Kavita Pal, Indraneel Mittra.

**Data curation:** Kavita Pal, Indraneel Mittra.

**Formal analysis:** Gorantla V. Raghuram, Kavita Pal, Indraneel Mittra.

**Funding acquisition:** Indraneel Mittra.

**Investigation:** Gorantla V. Raghuram, Gaurav Sriram, Afzal Khan, Ruchi Joshi, Vishalkumar Jadhav, Sushma Shinde, Alfina Shaikh, Bhagyeshri Rane, Harshada Kangne.

**Methodology:** Gorantla V. Raghuram, Kavita Pal, Indraneel Mittra.

**Project administration:** Gorantla V. Raghuram, Kavita Pal, Indraneel Mittra.

**Resources:** Indraneel Mittra.

**Software:** Kavita Pal.

**Supervision:** Kavita Pal, Indraneel Mittra.

**Validation:** Gorantla V. Raghuram, Kavita Pal, Indraneel Mittra.

**Visualization:** Gorantla V. Raghuram, Kavita Pal, Indraneel Mittra.

**Writing – original draft:** Indraneel Mittra.

**Writing – review & editing:** Indraneel Mittra.

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
