## [Decision Letter · Decision Letter 0]

9 Aug 2023

PONE-D-23-14930Therapeutic interventions on human breast cancer xenografts promote systemic dissemination of oncogenesPLOS ONE

Dear Dr. Mittra,

Thank you for submitting your manuscript to PLOS ONE. After careful consideration, we feel that it has merit but does not fully meet PLOS ONE’s publication criteria as it currently stands. Therefore, we invite you to submit a revised version of the manuscript that addresses the points raised during the review process.

I have carefully reviewed the reviewer's comment on the manuscript titled "Therapeutic Interventions on Human Breast Cancer Xenografts" and I appreciate the efforts you have invested in your research. However, there are some key concerns and suggestions that I believe will enhance the clarity and validity of your study. I recommend addressing these points before resubmitting your manuscript for further consideration:

Major Points:

**Clarification of Source of Signals:** To confirm that signals detected in brain sections indeed originate from cfChPs released from the xenografts and not due to metastasis, I recommend labeling MDA-MB-231 cells with a fluorescent protein and observing their presence in the brain. The absence of the fluorescent signal, along with the presence of human DNA sequences, would provide stronger evidence for the transfer of cfChPs from the tumor to recipient cells.

**High-Resolution Images: **Provide high-resolution versions of all figures and ensure that they are clear, crisp, and visually appealing. This will enable readers to better appreciate the details and nuances of your experimental results.

We look forward to receiving your revised manuscript.

Kind regards,

Vinod Vijayakurup, Ph.D.

Academic Editor

PLOS ONE

Journal Requirements:

 Whilst you may use any professional scientific editing service of your choice, PLOS has partnered with both American Journal Experts (AJE) and Editage to provide discounted services to PLOS authors. Both organizations have experience helping authors meet PLOS guidelines and can provide language editing, translation, manuscript formatting, and figure formatting to ensure your manuscript meets our submission guidelines. To take advantage of our partnership with AJE, visit the AJE website (http://aje.com/go/plos) for a 15% discount off AJE services. To take advantage of our partnership with Editage, visit the Editage website (www.editage.com) and enter referral code PLOSEDIT for a 15% discount off Editage services. If the PLOS editorial team finds any language issues in text that either AJE or Editage has edited, the service provider will re-edit the text for free.

 A clean copy of the edited manuscript (uploaded as the new *manuscript* file)”"

4. Please upload a new copy of Figure 2 as the detail is not clear. Please follow the link for more information: " ext-link-type="uri" xlink:type="simple">https://blogs.plos.org/plos/2019/06/looking-good-tips-for-creating-your-plos-figures-graphics/"
https://blogs.plos.org/plos/2019/06/looking-good-tips-for-creating-your-plos-figures-graphics/  

Reviewers' comments:

Reviewer's Responses to Questions

**Comments to the Author**

1. Is the manuscript technically sound, and do the data support the conclusions?

Reviewer #1: Yes

Reviewer #2: No

2. Has the statistical analysis been performed appropriately and rigorously? 

Reviewer #1: Yes

Reviewer #2: Yes

3. Have the authors made all data underlying the findings in their manuscript fully available?

Reviewer #1: Yes

Reviewer #2: Yes

4. Is the manuscript presented in an intelligible fashion and written in standard English?

Reviewer #1: Yes

Reviewer #2: Yes

5. Review Comments to the Author

Reviewer #1: The manuscript “Therapeutic interventions on human breast cancer xenografts promote systemic dissemination of oncogenes” is a small piece of work that explains probable reason for tumor relapse or metastasis after successfully treating the tumor with commonly adopted therapeutic modalities like chemotherapy, radiotherapy and surgery.

The authors hypothesize that therapeutic interventions lead to the dispersal of DNA fragments from the dying cancer cells, which carrying the oncoprotein. This is transported by blood to distant organ sites to induce tumor development and the use of agents that can deactivate these cell free DNA particles can lead to therapeutic benifits.

Authors have shown by immuno-FISH analysis, in mice xenograft brain sections, the co-localization of human DNA with HLA-ABC antigen, which is specific to human and different human oncogenes, after therapeutic interventions. The group has also suggested and shown that use of cell free chromatin particles deactivating agents along with the different therapies could inactive and prevent these cell free chromatin particles from inducing the toxic side effects.

One point I want to bring to notice is that the figures are blurred. Please enhance the figure quality.

Reviewer #2: The manuscript titled “Therapeutic interventions on human breast cancer xenografts promote systemic dissemination of oncogenes” describes a potential transfer of cell-free chromatin particles from tumor cells to normal cells in response to different treatment modalities. The novelty of the study is overall low and the experimental evidences for the interpretations made are weak.

Some of the points to be addressed are given below

Major points

1. The results do not explicitly show that the signals detected in brain sections are indeed from the cfChPs released from the xenograft. It can be caused due to metastasis of the tumor to brain. Ideally, MDA-MB-231 cells should have been labelled with a fluorescent protein to detect its presence in the brain. An absence of this signal with the presence of the human DNA sequences will confirm the transfer of cfChPs from the tumor to the recipient cells.

2. As a first step, the investigators could have assessed human circulating cell-free DNA in response to different treatment modalities compared to control (Xenograft only) and see if it correlates with a higher signal in the brain.

3. Another approach to confirm the transfer of cfChPs to the host recipient cell is by blocking endocytosis and see if there is reduction in the immuno-FISH signal. This can be first tested in vitro. ALternatively, try an approach similar to the one used in Mittra et al.,Cell Death and Discovery.2017(referred in the manuscript).

4. What is the rationale for choosing the proteins (p-EGFR,p-Akt,c-Myc,Raf,PDGFRA,HRAS,FGFR) listed in the results?

5. The results are not conclusive to show that DNA transferred from the tumor to the recipient cell has lead to the expression of the proteins (first paragraph of results). It could also be due to transfer of these proteins from the tumor cells via other routes like extracellular vesicles. Hence, this interpretation should be avoided.

Minor points

1. NF-ƘB is a transcription factor and not a cytokine. Correct this in the introduction.

2. Typing error in materials and methods under the heading "creation of xenografts"- MDA-MB-213 written instead of

231

6. PLOS authors have the option to publish the peer review history of their article (what does this mean?). If published, this will include your full peer review and any attached files.

Reviewer #1: No

Reviewer #2: No

---

## [Author Response · Author response to Decision Letter 0]

8 Sep 2023

Dear Dr. Vijayakurup

We have complied with comment no. 1 of reviewer 2 wherein a suggestion was made that we perform an experiment in which we inject fluorescently labelled MDA-MB-231 cells to prove that the fluorescent signals that we observed in brain cells of mice were not metastatic cells.

We have now performed such an experiment which is described in the “Methods” section; the results of which are described in the “Results” section, and the conclusions discussed in the “Discussion” section. This has required substantial changes to the manuscript (marked in red) and incorporation of a new figure marked as “Supplementary Fig. 3”.

5. Review Comments to the Author

Reviewer #1: The manuscript “Therapeutic interventions on human breast cancer xenografts promote systemic dissemination of oncogenes” is a small piece of work that explains probable reason for tumor relapse or metastasis after successfully treating the tumor with commonly adopted therapeutic modalities like chemotherapy, radiotherapy and surgery.

The authors hypothesize that therapeutic interventions lead to the dispersal of DNA fragments from the dying cancer cells, which carrying the oncoprotein. This is transported by blood to distant organ sites to induce tumor development and the use of agents that can deactivate these cell free DNA particles can lead to therapeutic benifits.

Authors have shown by immuno-FISH analysis, in mice xenograft brain sections, the co-localization of human DNA with HLA-ABC antigen, which is specific to human and different human oncogenes, after therapeutic interventions. The group has also suggested and shown that use of cell free chromatin particles deactivating agents along with the different therapies could inactive and prevent these cell free chromatin particles from inducing the toxic side effects.

One point I want to bring to notice is that the figures are blurred. Please enhance the figure quality.

The quality of the figures has now been enhanced. Thank you. 

Reviewer #2: The manuscript titled “Therapeutic interventions on human breast cancer xenografts promote systemic dissemination of oncogenes” describes a potential transfer of cell-free chromatin particles from tumor cells to normal cells in response to different treatment modalities. The novelty of the study is overall low and the experimental evidences for the interpretations made are weak.

Some of the points to be addressed are given below

Major points

1. The results do not explicitly show that the signals detected in brain sections are indeed from the cfChPs released from the xenograft. It can be caused due to metastasis of the tumor to brain. Ideally, MDA-MB-231 cells should have been labelled with a fluorescent protein to detect its presence in the brain. An absence of this signal with the presence of the human DNA sequences will confirm the transfer of cfChPs from the tumor to the recipient cells.

We thank the reviewer for making this valuable suggestion. We have now performed the experiment as suggested wherein we have dually fluorescently labeled MDA-MB-231 cells in their DNA with BrdU and in their histones with H2B-GFP by a method described by us earlier (reference no. 16 of the manuscript). The florescent cells were injected i.v. and brain cells were examined after 72 h. No intact fluorescent cells representing metastasis were seen in the brain. What we saw were dually labelled particle representing cfChPs. This confirms our hypothesis that cfChPs from dying xenograft cells go and accumulate in the brain cells and express the onco-proteins that they carry with them. 

This data has now been incorporated as Supplementary fig. 3 in the manuscript.

The above suggestion of the reviewer and the experimental procedure has required the addition of the following paragraphs:

1) pp. 10, lines 163-176 (Methods section)

2) pp. 12, lines 222-231 (Results section)

3) pp. 13, lines 239-249 (Discussion section) 

4) pp. 13-14, lines 251-258 (Discussion section)

This has also necessitated reorganizing the references. 

2. As a first step, the investigators could have assessed human circulating cell-free DNA in response to different treatment modalities compared to control (Xenograft only) and see if it correlates with a higher signal in the brain.

We humbly submit that in light of our response to reviewer’s point no. 1 above, this experiment is no longer required. 

3. Another approach to confirm the transfer of cfChPs to the host recipient cell is by blocking endocytosis and see if there is reduction in the immuno-FISH signal. This can be first tested in vitro. ALternatively, try an approach similar to the one used in Mittra et al.,Cell Death and Discovery.2017(referred in the manuscript).

We humbly submit that in light of our response to reviewer’s point no.1 above, this experiment is no longer required. 

4. What is the rationale for choosing the proteins (p-EGFR,p-Akt,c-Myc,Raf,PDGFRA,HRAS,FGFR) listed in the results?

There was no specific rationale except that these antibodies were readily available in our laboratory. The important point is that they are all onco-proteins. 

5. The results are not conclusive to show that DNA transferred from the tumor to the recipient cell has lead to the expression of the proteins (first paragraph of results). It could also be due to transfer of these proteins from the tumor cells via other routes like extracellular vesicles. Hence, this interpretation should be avoided.

Had the transfer of the onco-proteins from the tumor cells had occurred via other routes, such as via extracellular vesicles, they would not have been abrogated by treatment with the three cfChPs deactivating agents. 

Minor points

1. NF-ƘB is a transcription factor and not a cytokine. Correct this in the introduction.

We thank the reviewer for pointing this out. NFκB has now been appropriately specified (pp. 3, line 55).

2. Typing error in materials and methods under the heading "creation of xenografts"- MDA-MB-213 written instead of 231

Sorry for this typo. This has now been corrected (pp. 5, line 104).

---

## [Decision Letter · Decision Letter 1]

2 Nov 2023

PONE-D-23-14930R1Therapeutic interventions on human breast cancer xenografts promote systemic dissemination of oncogenesPLOS ONE

Dear Dr. Mittra,

Thank you for submitting your manuscript to PLOS ONE. After careful consideration, we feel that it has merit but does not fully meet PLOS ONE’s publication criteria as it currently stands. Therefore, we invite you to submit a revised version of the manuscript that addresses the points raised during the review process.

ACADEMIC EDITOR: Dear Authors,Please modify your work as per the suggesitons from our reviewers,

We look forward to receiving your revised manuscript.

Kind regards,

Chien-Feng Li, M.D., Ph.D.

Academic Editor

PLOS ONE

Journal Requirements:

Reviewers' comments:

Reviewer's Responses to Questions

**Comments to the Author**

1. If the authors have adequately addressed your comments raised in a previous round of review and you feel that this manuscript is now acceptable for publication, you may indicate that here to bypass the “Comments to the Author” section, enter your conflict of interest statement in the “Confidential to Editor” section, and submit your "Accept" recommendation.

Reviewer #1: (No Response)

Reviewer #2: All comments have been addressed

2. Is the manuscript technically sound, and do the data support the conclusions?

Reviewer #1: Yes

Reviewer #2: Partly

3. Has the statistical analysis been performed appropriately and rigorously? 

Reviewer #1: Yes

Reviewer #2: Yes

4. Have the authors made all data underlying the findings in their manuscript fully available?

Reviewer #1: Yes

Reviewer #2: Yes

5. Is the manuscript presented in an intelligible fashion and written in standard English?

Reviewer #1: Yes

Reviewer #2: Yes

6. Review Comments to the Author

Reviewer #1: The Figures are still not to the quality of publication I feel. Not much change has been made into the figure quality then the previous version of the manuscript.

Reviewer #2: 1. Supplementary figure.3 has to be replaced by a more high resolution image.

2. A query related to Point no.5 from the previous review- In the manuscript, the authors have shown increased c-Myc signal in the brain of tumor bearing animals with/without therapeutic intervention at varying intensity compared to control group where there is no signal. However, the antibody used here is reactive to mouse c-Myc and human c-Myc. Hence, the question of specificity still remains. Moreover, the possibility that the internalized DNA could be activating pathways like cGAS-STING cannot be ruled out. These pathways could be responsible for the elevated c-Myc signal. So if DNAase is treated, it will abrogate the entry of DNA and subsequent activation of the pathway. Hence, the results are not conclusive to show that DNA transferred from the tumor to the recipient cell has lead to the expression of the proteins. Either this sentence has to be omitted or the experiment has to be repeated with human specific c-Myc.

7. PLOS authors have the option to publish the peer review history of their article (what does this mean?). If published, this will include your full peer review and any attached files.

Reviewer #1: No

Reviewer #2: No

---

## [Author Response · Author response to Decision Letter 1]

10 Nov 2023

Review Comments to the Author

Reviewer #1: The Figures are still not to the quality of publication I feel. Not much change has been made into the figure quality then the previous version of the manuscript.

We are once again providing a high resolution images of all the figures and supplementary figures. We have used DPI Converter using the following link. All images have been enhanced to 600 dpi. 

https://convert.town/image-dpi

Reviewer #2: 1. Supplementary figure.3 has to be replaced by a more high resolution image.

This question has been answered above 

2. A query related to Point no.5 from the previous review- In the manuscript, the authors have shown increased c-Myc signal in the brain of tumor bearing animals with/without therapeutic intervention at varying intensity compared to control group where there is no signal. However, the antibody used here is reactive to mouse c-Myc and human c-Myc. Hence, the question of specificity still remains. 

We like to point out that we have used the Sigma Aldrich anti-human c-Myc monoclonal antibody M4439 which is clearly indicated in supplementary table 1.

This mAb is human specific as indicated in the company datasheet which is given as “supporting information”. The datasheet contains the following specificity statements:

Species reactivity: Human

Immunogen: Synthetic peptide of the human p62c-Myc protein.

Moreover, the possibility that the internalized DNA could be activating pathways like cGAS-STING cannot be ruled out. These pathways could be responsible for the elevated c-Myc signal. So if DNAase is treated, it will abrogate the entry of DNA and subsequent activation of the pathway. Hence, the results are not conclusive to show that DNA transferred from the tumor to the recipient cell has lead to the expression of the proteins. Either this sentence has to be omitted or the experiment has to be repeated with human specific c-Myc.

Since the mAb is human specific, the possibility that cGAS-STING pathway is responsible for the elevated c-Myc signal is ruled out.

 

---

## [Decision Letter · Decision Letter 2]

21 Dec 2023

PONE-D-23-14930R2Therapeutic interventions on human breast cancer xenografts promote systemic dissemination of oncogenesPLOS ONE

Dear Dr. Mittra,

Thank you for submitting your manuscript to PLOS ONE. After careful consideration, we feel that it has merit but does not fully meet PLOS ONE’s publication criteria as it currently stands. Therefore, we invite you to submit a revised version of the manuscript that addresses the points raised during the review process.

**ACADEMIC EDITOR: ** Dear Authors, Based on the reviewers' comments, I am inviting you to revise your work.

If applicable, we recommend that you deposit your laboratory protocols in protocols.io to enhance the reproducibility of your results. Protocols.io assigns your protocol its own identifier (DOI) so that it can be cited independently in the future. For instructions see: https://journals.plos.org/plosone/s/submission-guidelines#loc-laboratory-protocols. Additionally, PLOS ONE offers an option for publishing peer-reviewed Lab Protocol articles, which describe protocols hosted on protocols.io. Read more information on sharing protocols at https://plos.org/protocols?utm_medium=editorial-emailutm_source=authorlettersutm_campaign=protocols.

We look forward to receiving your revised manuscript.

Kind regards,

Chien-Feng Li, M.D., Ph.D.

Academic Editor

PLOS ONE

Reviewers' comments:

Reviewer's Responses to Questions

**Comments to the Author**

1. If the authors have adequately addressed your comments raised in a previous round of review and you feel that this manuscript is now acceptable for publication, you may indicate that here to bypass the “Comments to the Author” section, enter your conflict of interest statement in the “Confidential to Editor” section, and submit your "Accept" recommendation.

Reviewer #1: All comments have been addressed

Reviewer #2: All comments have been addressed

Reviewer #3: All comments have been addressed

2. Is the manuscript technically sound, and do the data support the conclusions?

Reviewer #1: Yes

Reviewer #2: Yes

Reviewer #3: Partly

3. Has the statistical analysis been performed appropriately and rigorously? 

Reviewer #1: Yes

Reviewer #2: Yes

Reviewer #3: Yes

4. Have the authors made all data underlying the findings in their manuscript fully available?

Reviewer #1: Yes

Reviewer #2: Yes

Reviewer #3: Yes

5. Is the manuscript presented in an intelligible fashion and written in standard English?

Reviewer #1: Yes

Reviewer #2: Yes

Reviewer #3: Yes

6. Review Comments to the Author

Reviewer #1: (No Response)

Reviewer #2: The authors have answered most of the queries raised by the reviewers. However, Figure 1b shows localization of PDGFRA, HRAS, c-Myc, c-Raf etc in the nucleus. While there are reports of transfer of membrane proteins to the nucleus, an exclusive localization to the nucleus is not common. Moreover, the significance of the result is not very clear. What is the relevance of localization of the chromatin particles with the mentioned oncogenic proteins in the nucleus? Please mention this in the manuscript.

Reviewer #3: Major concerns:

• Data presented is preliminary and requires extensive validation. For example, the data obtained by immuno-fluorescence and FISH analysis are only suggestive, further work is still needed to establish that it is cell free oncogenic DNA and proteins rather than MDA-MB-231 cells. One way to address this issue is by immunohistochemistry, which was not done. Moreover, the work described in supp. Figure 3 is difficult to interpret considering the difference in timeframe between the original work (figures 1 and 2), which lasted for 6 weeks and the work in supp. Figure 3, which lasted for only 72 hours.

• Using a xenograft mouse model to address this issue is problematic for many reasons, not least of which is that these are loose cells that can travel to different organs. A more suitable model is to work with a model of chemically- or virally-induce cancer. This way, the investigation starts with a localized cancer and oncogenic release of cell free DNA or proteins, rather than intact cells, can be tracked with greater confidence.

• The study is devoid of any mechanistic insight; experimentally speaking, how can cell free DNA and/or protein promote metastasis as per the model used in this study.

• There is a serious logical disconnect between the hypothesis put forward by the authors, which says that “therapeutic interventions may disseminate the disease 31 via medium of cfChPs released from therapy induced dying cancer cells” and the some of the validation work (results lines 219-228 supp. Figure 3) that was done to rule out that the signals detected in mouse brains were not metastatic cells?

Minor comments:

• There are no legends for figures 1 and 2

• The introduction is almost a repeat of the abstract

• The discussion is very short and barely addresses the findings in the context of what is already known regarding this phenomenon

7. PLOS authors have the option to publish the peer review history of their article (what does this mean?). If published, this will include your full peer review and any attached files.

Reviewer #1: No

Reviewer #2: No

Reviewer #3: **Yes: **Mawieh Hamad

---

## [Author Response · Author response to Decision Letter 2]

4 Jan 2024

Response to Reviewers

Reviewer #2: The authors have answered most of the queries raised by the reviewers. However, Figure 1b shows localization of PDGFRA, HRAS, c-Myc, c-Raf etc in the nucleus. While there are reports of transfer of membrane proteins to the nucleus, an exclusive localization to the nucleus is not common. Moreover, the significance of the result is not very clear. What is the relevance of localization of the chromatin particles with the mentioned oncogenic proteins in the nucleus? Please mention this in the manuscript.

We agree with the reviewer that there are several reports in the literature of transfer of membrane proteins to the nucleus (for example, Zuleger, N., et al. 2012. Cell. Mol. Life Sci., 69, 2205-2216 and Zheng, H.C. et al. 2022. Mol. Med. Reports, 25, 1-12.). However, in our case, the oncogenes contained within the cfChPs that had accumulated in brain cells were being directly expressed as proteins in their nuclei. The relevance of such a finding is that oncogenes carried by cfChPs via the blood stream to the brain can potentially transform the brain cells to form new cancers which would masquerade as metastasis. 

We have now addressed this issue on page 14, lines 282-287

Reviewer #3: Major concerns:

• Data presented is preliminary and requires extensive validation. For example, the data obtained by immuno-fluorescence and FISH analysis are only suggestive, further work is still needed to establish that it is cell free oncogenic DNA and proteins rather than MDA-MB-231 cells.

This issue was also raised earlier by the second reviewer who had suggested that we conduct the experiment which is now presented in supplementary figure 3. The latter shows that no intact intravenously injected MDA-MB-231 cells are present in the brain. The second reviewer is now satisfied with this experiment. 

We believe that the experiment described in Supplementary Figure 3 conclusively excludes the possibility that the fluorescence signals seen in Figure 1 represent intact MDA-MB-231. In this experiment we intravenously injected dually fluorescently labelled MDA-MB-231 cells and looked for intact fluorescent cells in the brain of mice. Intact cells would be expected to appear as large dually-labelled fluorescent signals which would co-localize with an entire DAPI nuclear signal. We did not find any such large co-localizing signals; instead we detected fluorescent particles representing cfChPs within the blue DAPI nuclear signals. This suggested that cfChPs from dying xenograft cells had been carried via the blood stream to accumulate in the brain cell nuclei. 

Additionally, the fact that the fluorescent signals could be markedly minimized by concurrent treatment with the three cfChPs deactivating enzymes strongly argues against the possibility that the fluorescent signals seen in brain cells represented intact MDA-MB-231 cells. Rather, they indicate that the vehicles that carried the c-Myc oncogene to brain cells were cfChPs released from dying xenograft cells which could be prevented by deactivating cfChPs.

We have now addressed the reviewer’s concerns in two separate paragraphs in the discussion section (pages 13-14, lines 259-281).

One way to address this issue is by immunohistochemistry, which was not done. 

The reviewer does not specify which antigen should be used as target for immunohistochemistry. After extensive search, we could not find a surface antigen which is unique to MDA-MB-231 cells and is not present in brain cells. For example, the epithelial cell surface marker ICAM-1 is also expressed by microglial cells in the brain.

Moreover, the work described in supp. Figure 3 is difficult to interpret considering the difference in timeframe between the original work (figures 1 and 2), which lasted for 6 weeks and the work in supp. Figure 3, which lasted for only 72 hours.

We agree that there is difference in the timeframe between figures 1 2 and supplementary figure 3. However, if no intact MDA-MB-231 cells in the brain were seen at 72 h (Supplementary Fig. 3), it is highly unlikely that the fluorescent signals seen at 6 weeks would represent intact MDA-MB-231 cells (Figures 1 and 2). In fact, the absence of intact cells at 72 h is in keeping with Fidler’s finding that 99% of injected cancer cells rapidly die within 24 h [11]. 

We have now addressed the issue of difference in timeframe in the discussion section (Page 14, Lines 267-273)

• Using a xenograft mouse model to address this issue is problematic for many reasons, not least of which is that these are loose cells that can travel to different organs. A more suitable model is to work with a model of chemically- or virally-induce cancer. This way, the investigation starts with a localized cancer and oncogenic release of cell free DNA or proteins, rather than intact cells, can be tracked with greater confidence.

We beg to disagree. The induction of chemically- or virally-injected tumours locally would represent an unnatural scenario and not represent the real world of human cancer (such as breast cancer). 

• The study is devoid of any mechanistic insight; experimentally speaking, how can cell free DNA and/or protein promote metastasis as per the model used in this study.

The mechanism that underpins our findings was described by us earlier wherein we had shown that cfChPs that circulate in blood of cancer patients can readily enter into cells of distant organs to damage their DNA and potentially transform them into new cancers which could appear as metastasis (Mittra, I., et al. (2015). J. biosci., 40, 91-111).

We have now addressed this mechanistic aspect of our finding in the manuscript (Page 13, Lines 255-258) 

• There is a serious logical disconnect between the hypothesis put forward by the authors, which says that “therapeutic interventions may disseminate the disease 31 via medium of cfChPs released from therapy induced dying cancer cells” and the some of the validation work (results lines 219-228 supp. Figure 3) that was done to rule out that the signals detected in mouse brains were not metastatic cells?

We thank the reviewer for pointing out this rather confusing interpretation of the hypothesis on our part. This is indeed a very important issue that requires detailed clarification.

The clarification that fluorescent signals seen in brain cells were cfChPs and not intact cells has been addressed above. Further clarification that cfChPs, rather than intact cells, are the potential agents that can transform target cells to generate new cancers that would masquerade as metastasis required clarification at several places

These clarifications have been made in several sentences in the abstract (marked in red); in several places in the introduction section (marked in red). The discussion section has been extensively re-written to emphasize this novel mechanism of metastasis which poses a formidable challenge to the existing model (marked in red). 

Minor comments:

• There are no legends for figures 1 and 2

There was a lot of confusion in the editorial office after we uploaded this manuscript, and were subsequently asked to send the manuscript by email. We have confirmed from our folder that we had sent the legends to figures as a separate file in our email to the editorial office. There seems to have been some error on part of the editorial office because of which the legends are missing. For the benefit of the reviewer we are reproducing below legends to figures 1 2. 

Legends to figures

Fig.1: Representative immuno-FISH images of FFPE sections of brains of mice bearing human breast cancer xenografts (without therapeutic interventions) showing co-localizing signals of human DNA and various human onco-proteins. a. Co-localizing signals of human DNA and human HLA-ABC. b. Co-localizing signals of human DNA and eight different human onco-proteins.

Fig. 2: Quantitative analyses depicted as histograms to illustrate that therapeutic interventions promote systemic dissemination of human DNA and human c-Myc onco-protein to mouse brain cells, and that these can be prevented by concurrent treatment with three different cfChPs deactivating agents. All groups had 4 mice each. a. Detection of human DNA by FISH b. Detection of human c-Myc onco-protein by immunofluorescence. Statistical comparison between the xenograft bearing group and the two control groups, and that between the xenograft bearing group and the three anti-cancer treatment groups (CT, RT and Sx) was done by two-tailed student t-test. * 0.05, ** 0.01, *** 0.001, **** 0.0001. Statistical comparison between the three anti-cancer treatment groups (CT, RT and Sx) and those additionally treated with the three cfChPs deactivating agents was done by One-way ANOVA. * 0.05, ** 0.01.

• The introduction is almost a repeat of the abstract

We thank the reviewer for pointing this out. We have now re-written the introduction to give it a different connotation (marked in red). 

• The discussion is very short and barely addresses the findings in the context of what is already known regarding this phenomenon

We thank the reviewer for making this observation. The discussion section has now been extensively re-written and enlarged (marked in red).

---

## [Editor Report · Decision Letter 3]

9 Jan 2024

PONE-D-23-14930R3Therapeutic interventions on human breast cancer xenografts promote systemic dissemination of oncogenesPLOS ONE

Dear Dr. Mittra,

Thank you for submitting your manuscript to PLOS ONE. After careful consideration, we feel that it has merit but does not fully meet PLOS ONE’s publication criteria as it currently stands. Therefore, we invite you to submit a revised version of the manuscript that addresses the points raised during the review process.

If applicable, we recommend that you deposit your laboratory protocols in protocols.io to enhance the reproducibility of your results. Protocols.io assigns your protocol its own identifier (DOI) so that it can be cited independently in the future. For instructions see: https://journals.plos.org/plosone/s/submission-guidelines#loc-laboratory-protocols. Additionally, PLOS ONE offers an option for publishing peer-reviewed Lab Protocol articles, which describe protocols hosted on protocols.io. Read more information on sharing protocols at https://plos.org/protocols?utm_medium=editorial-emailutm_source=authorlettersutm_campaign=protocols.

We look forward to receiving your revised manuscript.

Kind regards,

Chien-Feng Li, M.D., Ph.D.

Academic Editor

PLOS ONE

Additional Editor Comments:

**ACADEMIC EDITOR:** Dear authors, I am inviting you to revise your work again as per the reviewers' comments.

---

## [Author Response · Author response to Decision Letter 3]

10 Jan 2024

Revision 3

This question actually pertains to review no. 2 to which we had already responded.

3. The authors have answered most of the queries raised by the reviewers. However, I have noticed that in Figure 1b shows localization of PDGFRA, HRAS, c-Myc, c-Raf etc in the nucleus. While there are reports of transfer of membrane proteins to the nucleus, an exclusive localization to the nucleus is not common. Moreover, the significance of the result is not very clear. What is the relevance of localization of the chromatin particles with the mentioned oncogenic proteins in the nucleus? Please mention this in the manuscript.

We agree with the reviewer that there are several reports in the literature of transfer of membrane proteins to the nucleus (for example, Zuleger, N., et al. 2012. Cell. Mol. Life Sci., 69, 2205-2216 and Zheng, H.C. et al. 2022. Mol. Med. Reports, 25, 1-12.). However, in our case, the oncogenes contained within the cfChPs that had accumulated in brain cells were being directly expressed as proteins in their nuclei. The relevance of such a finding is that oncogenes carried by cfChPs via the blood stream to the brain can potentially transform the brain cells to form new cancers which would masquerade as metastasis. 

We have now addressed this issue on page 14, lines 282-287

Revision 2 

The below question is the only new question this time round

1. Supplementary figure.3 has to be replaced by a more high resolution image.

We have already provided high resolution image of Supplementary Figure 3. We used the DPI Converter using the following link. All images have been enhanced to 600 dpi. 

https://convert.town/image-dpi

Perhaps the reviewer has been provided with a PDF version of the image.

Nonetheless, we are again uploading the same high resolution image.

The below question actually pertains to review no. 1 to which we had already responded.

2. Point no.5 from the previous review (reviewer 2) has not been addressed properly. In the manuscript, the authors have shown increased c-Myc signal in the brain of tumor bearing animals with/without therapeutic intervention at varying intensity compared to control group where there is no signal. However, the antibody used here is reactive to mouse c-Myc and human c-Myc. Hence, the question of specificity still remains. 

We like to point out that we have used the Sigma Aldrich anti-human c-Myc monoclonal antibody M4439 which is clearly indicated in supplementary table 1.

This mAb is human specific as indicated in the company datasheet which is given as “supporting information”. The datasheet contains the following specificity statements:

Species reactivity: Human

Immunogen: Synthetic peptide of the human p62c-Myc protein.

Moreover, the possibility that the internalized DNA could be activating pathways like cGAS-STING cannot be ruled out. These pathways could be responsible for the elevated c-Myc signal. So if DNAase is treated, it will abrogate the entry of DNA and subsequent activation of the pathway. Hence, the results are not conclusive to show that DNA transferred from the tumor to the recipient cell has lead to the expression of the proteins. Either this sentence has to be omitted or the experiment has to be repeated with human specific c-Myc. 

Since the mAb is human specific, the possibility that cGAS-STING pathway is responsible for the elevated c-Myc signal is ruled out.

---

## [Editor Report · Decision Letter 4]

17 Jan 2024

Therapeutic interventions on human breast cancer xenografts promote systemic dissemination of oncogenes

PONE-D-23-14930R4

Dear Dr. Mittra,

We’re pleased to inform you that your manuscript has been judged scientifically suitable for publication and will be formally accepted for publication once it meets all outstanding technical requirements.

Kind regards,

Chien-Feng Li, M.D., Ph.D.

Academic Editor

PLOS ONE
---

## [Editor Report · Acceptance letter]

2 Feb 2024

PONE-D-23-14930R4 

PLOS ONE

Dear Dr. Mittra, 

I'm pleased to inform you that your manuscript has been deemed suitable for publication in PLOS ONE. Congratulations! Your manuscript is now being handed over to our production team.

Kind regards, 

on behalf of

Dr. Chien-Feng Li 

Academic Editor

PLOS ONE